# An antioxidant screen identifies ascorbic acid for prevention of light-induced mitotic prolongation in live cell imaging

Tomoki Harada[1,3], Shoji Hata [1,2,3✉], Rioka Takagi[1], Takuma Komori[1], Masamitsu Fukuyama[1], Takumi Chinen[1] & Daiju Kitagawa [1✉]

Phototoxicity is an important issue in fluorescence live imaging of light-sensitive cellular processes such as mitosis. Among several approaches to reduce phototoxicity, the addition of antioxidants to the media has been used as a simple method. Here, we analyzed the impact of phototoxicity on the mitotic progression in fluorescence live imaging of human cells and performed a screen to identify the most efficient antioxidative agents that reduce it. Quantitative analysis shows that high amounts of light illumination cause various mitotic defects such as prolonged mitosis and delays of chromosome alignment and centrosome separation. Among several antioxidants, our screen reveals that ascorbic acid significantly alleviates these phototoxic effects in mitosis. Furthermore, we demonstrate that adding ascorbic acid to the media enables fluorescence imaging of mitotic events at very high temporal resolution without obvious photodamage. Thus, this study provides an optimal method to effectively reduce the phototoxic effects in fluorescence live cell imaging.

[1] Department of Physiological Chemistry, Graduate School of Pharmaceutical Sciences, The University of Tokyo, Bunkyo, Tokyo, Japan. [2] Precursory Research for Embryonic Science and Technology (PRESTO) Program, Japan Science and Technology Agency, Honcho Kawaguchi, Saitama, Japan. [3]These authors contributed equally: Tomoki Harada, Shoji Hata. ✉email: s.hata@mol.f.u-tokyo.ac.jp; dkitagawa@mol.f.u-tokyo.ac.jp

Live cell imaging is a powerful technique for studying dynamic cellular processes. Recent advances in fluorescence imaging have enabled us to visualize molecular dynamics with high spatiotemporal resolution[1]. However, the application of such high-resolution imaging on living cells is limited due to phototoxicity caused by high-intensity or prolonged sample illumination. Phototoxicity occurs through the light-induced formation of reactive oxygen species (ROS)[2,3], which react with numerous cellular components and disrupt their structures and functions. Severe oxidization results in cellular abnormalities such as membrane blebbing and subsequent cell death[4,5]. Even without apparent changes in cellular morphology, high amounts of light illumination can cause substantial cell dysfunction, such as abnormal intracellular calcium homeostasis[6–9]. Therefore, reducing phototoxicity in fluorescence live cell imaging is an important factor to accurately capture cellular processes with high spatiotemporal resolution and to avoid misinterpretation of experimental results.

Mitosis is a series of dynamic processes through which a parent cell divides into two genetically identical daughter cells[10]. Live imaging of these mitotic events however is not trivial, as they are known to be particularly sensitive to phototoxicity. Excitation light, especially at a wavelength of 488 nm, has been shown to inhibit mitotic entry in a dose-dependent manner[11] and to cause prolongation of mitotic duration[12]. Since cells round up and increase their height during mitosis[13], three-dimensional (3D) imaging that covers the entire cell over a wide z-axis range is necessary to capture the spatial dynamics of all relevant cellular components in this process. Furthermore, high temporal resolution is required to record the highly dynamic and complex nature of mitosis, which is completed in less than an hour. However, 3D time-lapse imaging with short intervals requires a very high frequency of excitation light exposure, which can induce phototoxicity in living cells.

Several approaches have been proposed to decrease photodamage in fluorescence imaging of living cells, including the use of advanced microscopes with low phototoxicity such as light-sheet microscopes[5]. However, these microscopes are often expensive and difficult to implement. The use of excitation light at a longer wavelength can also reduce cellular phototoxicity, although the effective brightness of fluorescent proteins excited at longer wavelengths, such as near-infrared fluorescent proteins, is much lower than that of green or red fluorescent proteins in mammalian cells[3]. Adding antioxidants to the live cell imaging buffer is another approach to reduce cellular phototoxicity by scavenging ROS and limiting oxidative stress upon light illumination[3]. The antioxidants sodium pyruvate and Trolox have been shown to protect cells from light-induced cell death[14] and G2 arrest[11], respectively. Although the use of antioxidants in the live cell imaging buffer is a simple and effective approach to generally reduce photodamage, the optimal antioxidative agents for the specific aim to accurately capture mitotic dynamics in fluorescence microscopy remain unclear.

In this study, we performed a screen to identify antioxidants that are capable of alleviating light-induced mitotic abnormalities in fluorescence live imaging. Among several compounds, ascorbic acid, also known as vitamin C, significantly reduces photodamage in mitotic cells without showing indications of cytotoxic side-effects. We demonstrate that at the appropriate concentrations required to reduce mitotic phototoxicity, ascorbic acid does not perturb cell survival, the accurate segregation of the chromosomes, or cell-cycle progression. Moreover, the addition of ascorbic acid to the imaging media enables time-lapse 3D imaging of mitotic processes at very short temporal intervals, which has been so far difficult to achieve without obvious photodamage. Therefore, this study provides an effective solution for reducing the mitotic phototoxicity and demonstrates its application for fluorescence imaging of mitotic dynamics at very high spatio-temporal resolution in living cells.

## Results

**High-light illumination causes abnormal prolongation of mitosis and delays chromosome alignment and centrosome separation in live cell 3D imaging.** We first investigated the vulnerability of the mitotic processes to phototoxicity in fluorescence live cell imaging of RPE1 cells, which are one of the most widely used non-transformed human diploid cells. To this end, we used RPE1 cells stably expressing mNeonGreen (mNG)-fused Histone H2B and mRuby2-fused γ-tubulin to visualize the dynamics of chromosomes and centrosomes, respectively[15], which allowed us to observe key events in the mitotic progression. To analyze photodamage in mitosis, we set two different conditions of excitation laser light illumination referred to as low and high conditions (Fig. 1a). In the high condition, we used 488 nm wavelength excitation light with >6 times higher laser power and 4 times longer exposure time compared to the low condition. The setting for the 561 nm wavelength excitation light was the same in both. Under each condition, z-stack images (21 planes with 1 μm z-steps) of the RPE1 cells were taken using a spinning disk confocal microscope in time-lapse at 3-minutes intervals for a total duration of 12 h.

Using this setup, we observed that the timing of chromosome segregation was delayed in the mitotic cells exposed to high illumination (Fig. 1b and Supplementary Movie 1). To compare mitotic duration, the time from nuclear envelope breakdown (NEBD) to chromosome segregation was measured for each condition. As in previous reports, the duration of mitosis in RPE1 cells was approximately 20 min in the low light condition[15,16] (Fig. 1c). In contrast, cells exposed to the high-light illumination took a significantly longer time to initiate chromosome segregation after NEBD. This data is consistent with the previous study showing that strong illumination of blue light prolongs the duration of mitosis in RPE1 cells[12]. Further analysis of chromosome dynamics revealed that the time from NEBD to chromosome alignment at the metaphase plate was prolonged in the high condition (Fig. 1d, e), indicating that chromosome congression is particularly sensitive to illumination with blue light. We also found that centrosome behavior before NEBD was affected in the high condition (Fig. 1b and Supplementary Movie 1). At the onset of mitosis, the two centrosomes are separated from each other at precise timing to initiate the formation of a bipolar spindle[15,17]. When NEBD was used as a reference for mitotic events, centrosomes were separated at −29.7 min and −21.8 min from NEBD on average in the low and high conditions, respectively (Fig. 1b, f). This data shows that the timing of centrosome separation is delayed by high-light illumination. It should be noted that, even under the high condition, neither cell death nor other severe abnormalities were observed, suggesting that these mitotic events are specifically sensitive to phototoxicity. Taken together, these results indicate that high amounts of blue light illumination in live cell imaging cause various mitotic abnormalities.

**Shortening the total acquisition time by cell-cycle synchronization does not alleviate the mitotic phototoxicity in live cell imaging.** During the live cell imaging, cells entered mitosis at various time points due to the asynchronous culturing condition. Given that the accumulation of ROS causes cellular abnormalities[18], cells that enter mitosis at later time points during the fluorescence imaging are expected to exhibit stronger mitotic phototoxicity than those that initiate division at earlier time points. In fact, a positive correlation between the duration of

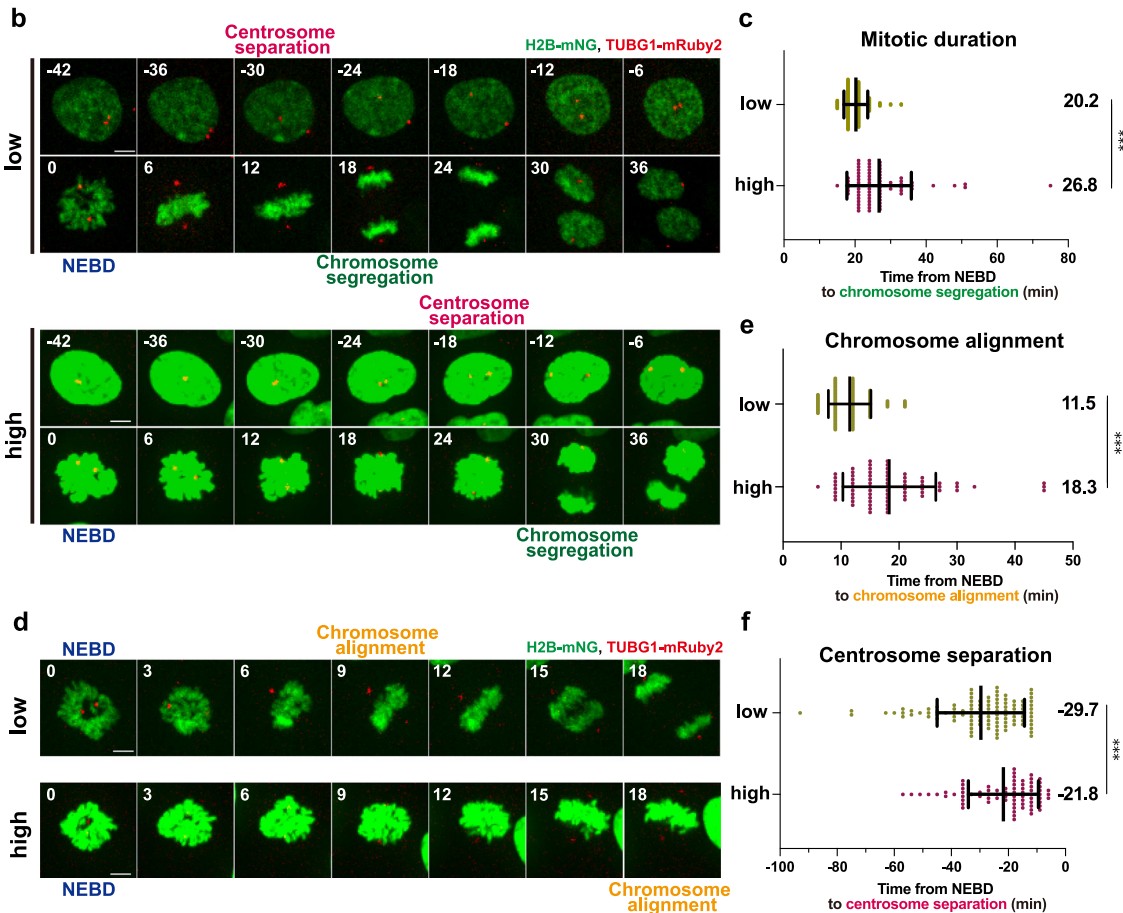

| Light condition | Excitation | Laser power | Exposure time | Z-stack | Intervals |
|---|---|---|---|---|---|
| low | 488 nm | 15% (0.70 W/cm²) | 50 msec | 1 μm step × 21 slices | 3 min |
| | 561 nm | 20% (0.91 W/cm²) | 100 msec | | |
| high | 488 nm | 100% (4.67 W/cm²) | 200 msec | 1 μm step × 21 slices | 3 min |
| | 561 nm | 20% (0.91 W/cm²) | 100 msec | | |

**Fig. 1 Intensive light exposure induces mitotic prolongation and delays chromosome alignment and centrosome separation in live cell imaging. a** The imaging conditions used for fluorescence live cell imaging. Apart from the variable intensity and exposure time of the 488-nm excitation light, all other settings are the same between the low and high conditions. **b** Time-lapse imaging of mitotic RPE1 cells stably expressing H2B-mNG and TUBG1-mRuby2 in the low and high conditions (3-min imaging intervals). Representative still images with different settings of brightness and contrast are shown due to the intense illumination in the high condition. T = 0 is designated as the time point of nuclear envelope breakdown (NEBD, time shown in min). The timing of centrosome separation and chromosome segregation are indicated. Scale bar, 5 μm. **c**, Quantification of mitotic duration from **b**. The time from NEBD to chromosome segregation was measured. n > 70 cells from three independent experiments. **d** Time-lapse images from NEBD to chromosome alignment in the low and high conditions (3-min intervals) from the same samples depicted in **b**. Representative still images are shown as in **b**. **e** Quantification of the time required for chromosome alignment from **d**. The time from NEBD to chromosome alignment at the metaphase plate was measured. n > 70 cells from three independent experiments. **f** Quantification of the timing of centrosome separation from **b**. The time from centrosome separation (defined by >4 μm inter-centrosomal distance) to NEBD was measured. n > 70 cells from four independent experiments. In **c**, **e**, and **f**, data are mean ± S.D., and P values were calculated by Mann-Whitney U-test. ***P < 0.001.

mitosis and the timing of mitotic entry after the start of live imaging was observed under the high-light illumination (Fig. 2a, b and Supplementary Movie 2). In particular, the cells that entered mitosis in the first half of the acquisition (within 6 h from the onset of imaging) showed a shorter duration of mitosis compared to those in the latter half (6–12 h), and even those in the total imaging time (0–6 and 6–12 h are combined) (Fig. 2c).

This data suggests that the extent to which cells are exposed to blue light before entering mitosis is an important factor in the occurrence of the mitotic phototoxicity and that reducing the pre-mitotic light exposure time would alleviate the phototoxicity.

Whereas shortening the total acquisition time of live imaging is one approach to reduce the pre-mitotic light exposure time of cells, this limits the number of mitotic cells that could be recorded

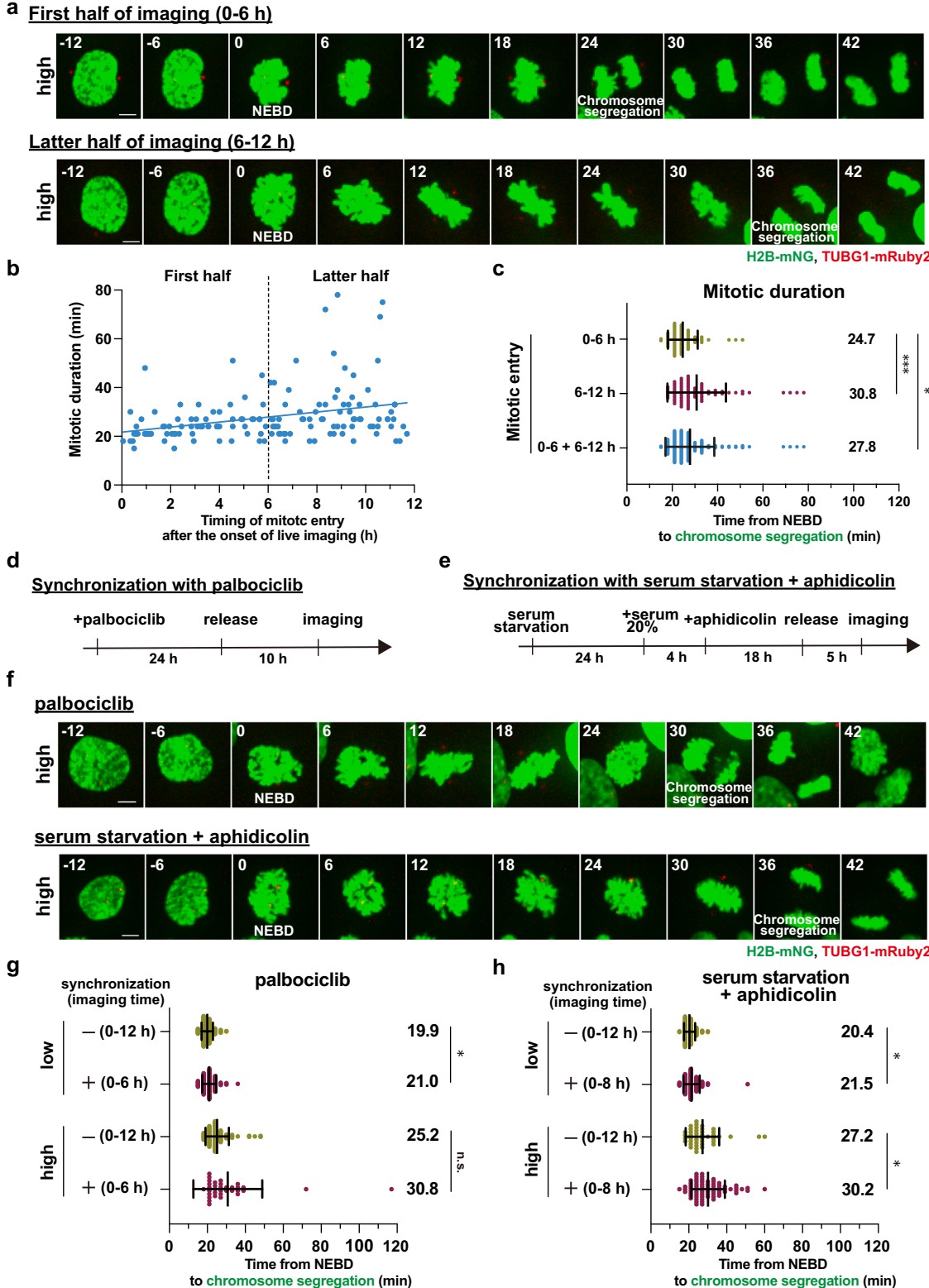

in a single experiment. In order to increase the number of the cells that enter mitosis within a narrow time window, we examined two different cell-cycle synchronization methods to enrich pre-mitotic cells prior to live cell imaging (Fig. 2d, e). The cell cycle progression of RPE1 cells has been shown to be highly synchronized after release from G1 arrest induced with palbociclib, a CDK4/6 inhibitor[19]. Since we observed that the cells synchronized with palbociclib began to enter mitosis at

13–14 h after the G1 release (Supplementary Fig. 1a, b), live cell imaging was started 10 h after the palbociclib washout. Unexpectedly, even though the cell-cycle synchronization allowed us to shorten the total acquisition time (6 h), compared to the control unsynchronous culture (12 h), this approach did not suppress the light-induced mitotic prolongation (Fig. 2f, g and Supplementary Movie 2). Furthermore, even under the low-light illumination, the synchronized cells showed a slight prolongation of mitosis. This

**Fig. 2 Reducing the total acquisition time by cell-cycle synchronization does not alleviate the light-induced mitotic prolongation. a** Time-lapse imaging of mitotic cells under the high condition (3-min intervals). Representative still images in the first and latter halves of the total acquisition time are shown. T = 0 is designated as NEBD (time shown in min). Scale bar, 5 μm. **b** The correlation between the time point of mitotic entry after the start of live imaging (x-axis, hours) and mitotic duration (y-axis, minutes) from **a**. NEBD was defined as the timing of mitotic entry. The regression line is shown. The vertical dotted line indicates the boundary of the first and latter 6 h-halves after the onset of live imaging. n = 150 cells from six independent experiments. **c** Quantification of mitotic duration from **a**. **d** Experimental procedure for cell-cycle synchronization with palbociclib treatment. **e** Experimental procedure for cell-cycle synchronization with serum starvation plus aphidicolin treatment. **f** Time-lapse imaging of mitotic cells synchronized with 300 nM palbociclib or with serum starvation plus 1 μM aphidicolin under the high illumination conditions (3-min intervals). Representative still images are shown. T = 0 is designated as NEBD (time shown in min). Scale bar, 5 μm. **g** Quantification of mitotic duration in the indicated conditions for the synchronization with palbociclib from **f**. In the control and synchronized cells, live imaging was performed for 12 h and 6 h, respectively. n > 30 cells from three independent experiments. **h** Quantification of mitotic duration in the indicated conditions for the synchronization with serum starvation plus aphidicolin treatment from **f**. In the control and synchronized cells, live imaging was performed for 12 h and 8 h, respectively. n > 40 cells from three independent experiments. In **c, g**, and **h**, data are mean ± S.D., and P values were calculated by Mann-Whitney U-test. *P < 0.05, **P < 0.01, ***P < 0.001. n.s.: not significant.

result suggests that cell-cycle synchronization with palbociclib itself is potentially harmful to the cells.

We therefore explored another cell-cycle synchronization method, in which RPE1 cells were first arrested in G0 by serum starvation, released, then again arrested in G1/S with a low concentration of aphidicolin, and finally released to undergo mitosis in an accurately synchronized manner[20,21]. As the synchronized cells were entering mitosis from 8–9 h after the G1/S release, live cell imaging was started 5 h after the washout of aphidicolin (Fig. 2e and Supplementary Fig. 1c, d). Quantification analysis revealed that this synchronization approach also does not alleviate and even enhances the light-induced mitotic prolongation (Fig. 2f, h and Supplementary Movie 2). Similar to the case of palbociclib treatment, cell-cycle synchronization with this two-step method resulted in a slight prolongation of mitosis even under the low-light illumination. Taken together, these data indicate that reducing the pre-mitotic light exposure time by cell-cycle synchronization does not alleviate the mitotic phototoxicity in fluorescence imaging with RPE1 cells, likely due to other toxic side-effects contributed by the chemical treatments.

**An antioxidant screen identifies ascorbic acid as a potent agent to prevent phototoxicity in mitosis.** In order to reduce the phototoxicity in mitosis during fluorescence live cell imaging, we next examined the approach of adding antioxidants to the imaging medium. To identify the antioxidants that specifically protect mitotic cells from photodamage by high-light illumination, we performed a small screen of antioxidative agents known to reduce cellular phototoxicity[9,11,14,22–24]. Each antioxidant was examined in two different concentrations, the lower of which was chosen based on what is commonly used in live cell imaging experiments. Among the six compounds tested, we found that neither Trolox nor zeaxanthin, sodium pyruvate, α-tocopherol, Rutin, or N-Acetyl-L-cysteine (NAC) was able to prevent the light-induced mitotic prolongation (Fig. 3a–g and Supplementary Movie 3). This is surprising since Trolox has been shown previously to suppress the delay of mitotic entry caused by high-light illumination in RPE1 cells[11] (Fig. 3a, b and Supplementary Movie 3). In contrast, the addition of ascorbic acid to the imaging medium at 500 μM almost completely restored the duration of mitosis in the high-light condition to the values observed under low-light illumination (from 28.8 min to 19.8 min on average, Fig. 3a, h and Supplementary Movie 3). Additionally, the mitotic duration was shorter in the presence of 500 μM ascorbic acid than that of 250 μM, suggesting a dose-dependent photoprotective effect of this antioxidant. Moreover, we observed that the treatment with 500 μM ascorbic acid, but not the other antioxidants, significantly suppressed the delays in chromosome congression and centrosome separation shown to be induced by high amounts of light exposure (Fig. 3i, j, Supplementary Fig. 2a-g and Supplementary Movie 3).

Fluorescent proteins are known to generate ROS when they absorb photons, thereby damaging cellular components in their vicinity[25]. We therefore investigated whether ascorbic acid exhibits a photoprotective effect against blue light illumination, regardless of the localization of green fluorescent proteins. To accomplish this, we generated RPE1 cells that stably express mNG-fused TUBB5, localizing to microtubules, along with mScarlet-fused H2B. Similar to the case of chromatin-localized H2B-mNG, the addition of ascorbic acid at 500 μM completely alleviated the prolongation of mitosis in RPE1 cells expressing TUBB5-mNG caused by high amounts of blue light illumination (Supplementary Fig. 3a, b). This indicates that the photoprotective effect of ascorbic acid is independent of the localization of fluorescent proteins.

To verify the photoprotective effect of ascorbic acid in different cell lines, we evaluated it in a human diploid fibroblast cell line BJ-5ta. Similar to the RPE1 cells, the duration of mitosis in BJ-5ta cells expressing H2B-mNG and TUBG1-mRuby2 was prolonged under the high irradiation of blue light, and this prolongation was significantly reduced by treatment with 500 μM ascorbic acid (Supplementary Fig. 3c–e). These results revealed that ascorbic acid, unlike other commonly used antioxidants, is capable of specifically protecting mitotic cells from phototoxicity.

**Ascorbic acid is not cytotoxic at the concentration required for prevention of mitotic phototoxicity.** We next examined whether the concentration of ascorbic acid exerting the photoprotective effect could be generally harmful to the cells. To this end, RPE1 cells were treated with various concentrations of ascorbic acid for 72 h and cell viability was subsequently determined by MTT assay for each condition (Fig. 4a–c). A dose-dependent curve of ascorbic acid on cell viability was plotted and the 50% inhibitory concentration value (IC50) was calculated as 3.38 mM. Whereas ascorbic acid concentrations above 1 mM reduced the viability of RPE1 cells, lower concentrations had no effect. The pH of the medium measured inside the 5% $CO_2$ incubator was almost identical between the control medium (pH 7.94 ± 0.02) and the medium containing 500 μM ascorbic acid (pH 7.97 ± 0.01), a concentration sufficient to suppress the mitotic phototoxicity. Furthermore, fluorescence live cell imaging in the low-light illumination condition confirmed that the accuracy of chromosome segregation was not affected in the presence of ascorbic acid at 500 μM (Fig. 4d, e). Moreover, the cell-cycle analysis assessed by flow cytometry revealed no significant differences in the proportion of cell-cycle phases between conditions with and without ascorbic acid at 500 μM (Fig. 4f, g and Supplementary Fig. 4). The total duration of cell cycle was also not affected with 500 μM of ascorbic acid (Fig. 4h, i), suggesting that the optimal dose of ascorbic acid for prevention of phototoxicity does not affect cell-cycle progression. Thus, these data indicate

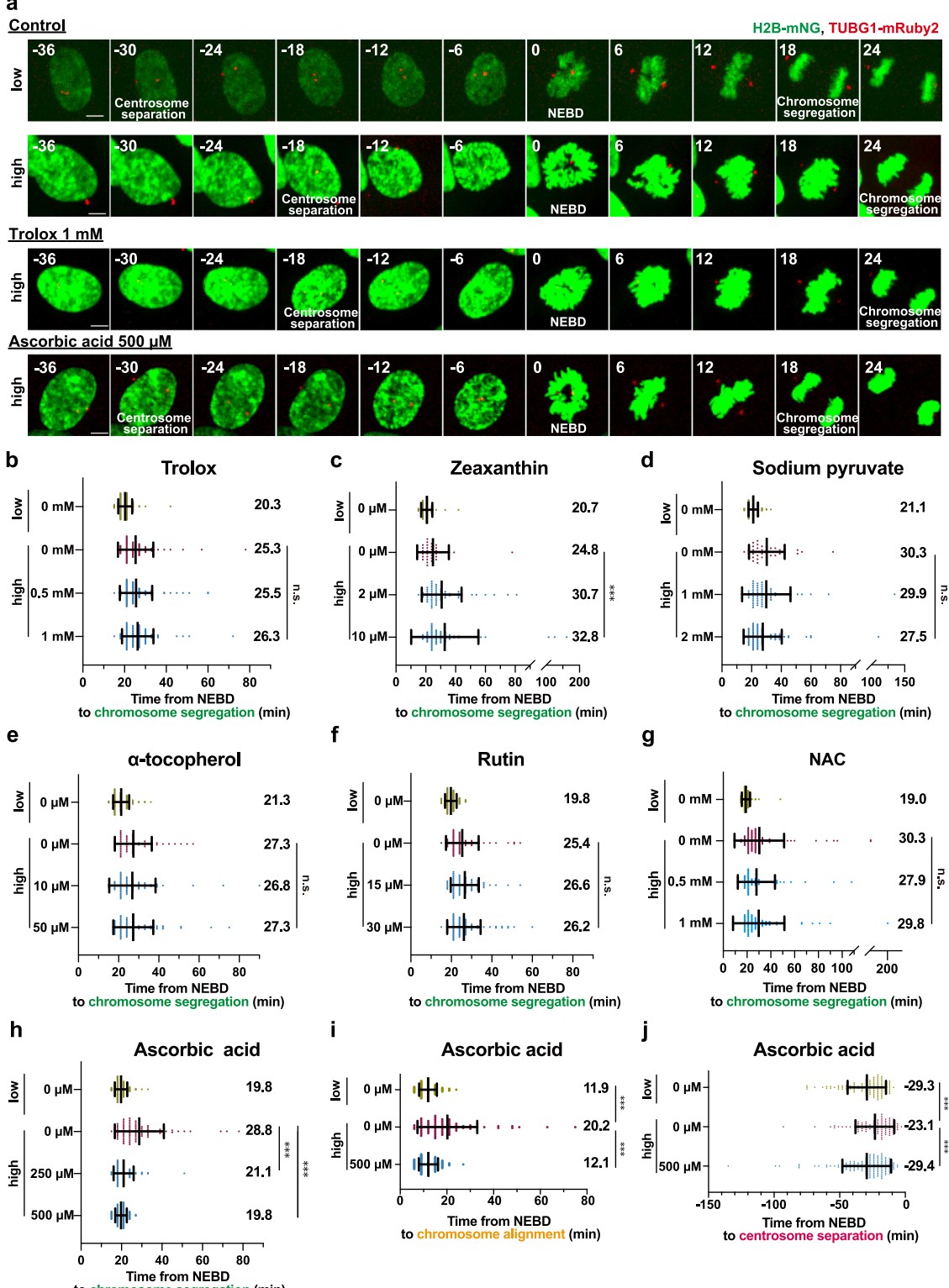

**Fig. 3 An antioxidant screen identifies ascorbic acid for reducing phototoxicity to mitotic cells. a** Time-lapse imaging of mitotic cells in the presence of antioxidants. Representative still images of control cells or cells imaged in the presence of 1 mM Trolox, or 500 µM ascorbic acid, are shown (3-min intervals). Images with different settings of brightness and contrast are shown. T = 0 is designated as NEBD (time shown in min). Scale bar, 5 µm. **b–h** Quantification of mitotic durations in the presence of the following antioxidants from **a**. **b**: Trolox, **c**: Zeaxanthin, **d**: Sodium pyruvate, **e**: α-tocopherol, **f**: Rutin, **g**: NAC, and **h**: Ascorbic acid. n > 30 cells from three independent experiments. **i**, Quantification of the time required for chromosome alignment in the absence and the presence of 500 µM ascorbic acid from **a**. The time from NEBD to chromosome alignment at the metaphase plate was measured. n > 70 cells from three independent experiments. **j**, Quantification of the timing of centrosome separation in the absence and the presence of 500 µM ascorbic acid from **a**. The time from centrosome separation (>4 µm inter-centrosomal distance) to NEBD was measured. n > 90 cells from five independent experiments. In **b–j**, data are mean ± S.D., and P values were calculated by Mann-Whitney U-test. ***P < 0.001, n.s.: not significant.

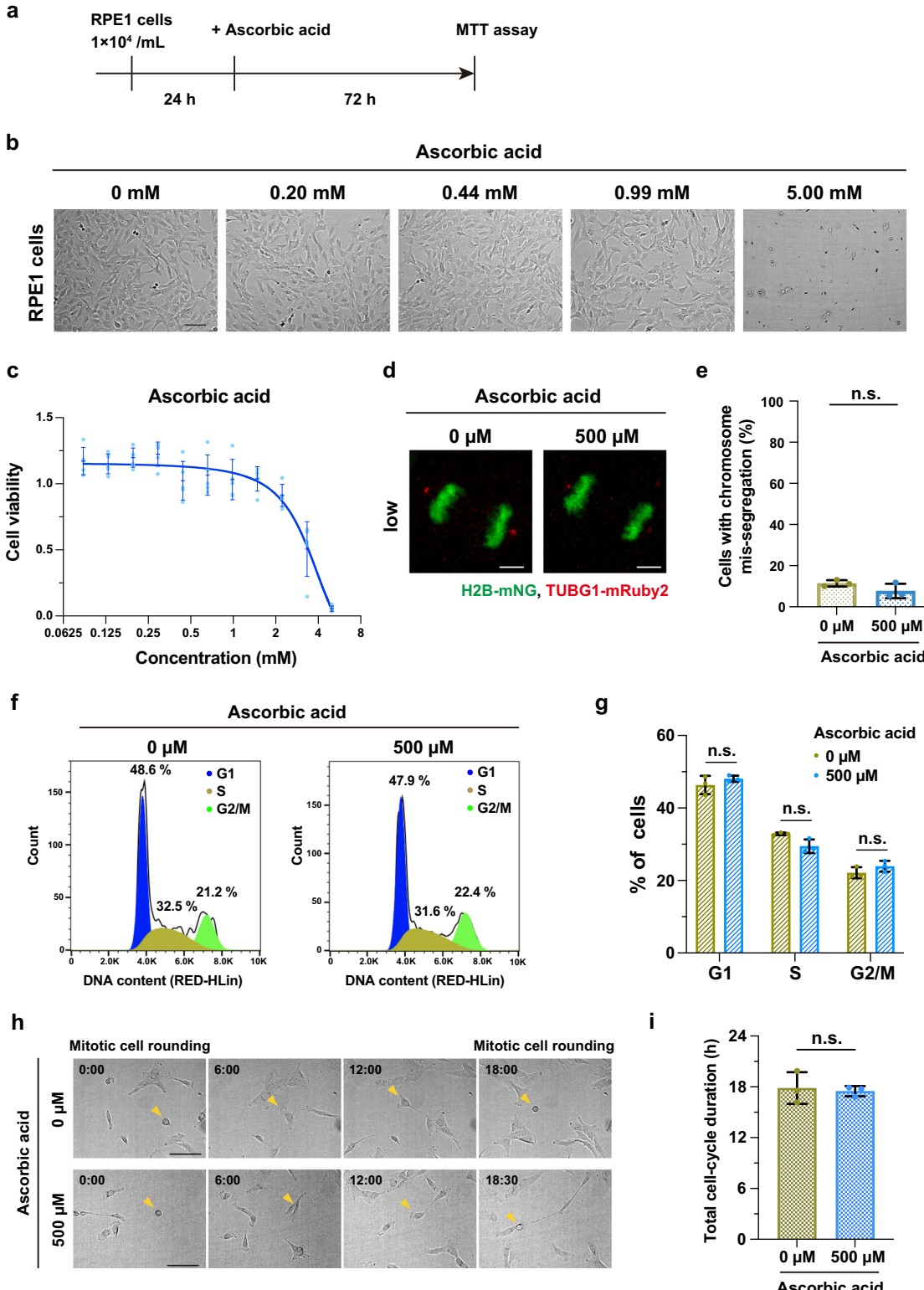

that the phototoxicity to mitosis can be safely suppressed by the use of ascorbic acid at a non-cytotoxic dose.

**The presence of ascorbic acid allows very high temporal resolution imaging of mitosis without obvious photodamage.** High temporal resolution imaging is necessary to capture the rapid dynamics of mitotic processes. However, shortening the interval between two frames in time-lapse imaging results in more

intensive light exposure of the cell sample, which in turn leads to higher phototoxicity. We therefore hypothesized that adding ascorbic acid to the imaging media could alleviate the severe phototoxicity in mitosis caused by the shortening of imaging intervals. To test this idea, we set a new imaging condition with 30-sec intervals, which is equivalent in light intensity to the low condition with 3-min intervals (Fig. 5a). Whereas the short-interval condition allowed us to capture the fine movement of chromosomes in mitosis, it resulted in a significantly longer

**Fig. 4 Ascorbic acid at concentrations adequate to prevent the mitotic phototoxicity is not cytotoxic. a** Experimental procedure for cell viability assay of RPE1 cells in the presence of various concentrations of ascorbic acid. **b** Representative bright-field images for the indicated conditions. Scale bar, 100 μm. **c** Quantification of dose-dependent effect of ascorbic acid on cell viability assessed by MTT assay. Cell viability was normalized with 0 mM ascorbic acid. $n = 5$ biological replicates for each concentration. Values are given as mean ± S.D. and the interpolation curve is shown. **d** Chromosome segregation of RPE1 cells stably expressing H2B-mNG and TUBG1-mRuby2 at the indicated concentrations of ascorbic acid in the low-light condition. Representative images of anaphase cells are shown. Scale bar, 5 μm. **e** Quantification of the percentage of chromosome mis-segregation from $n > 90$ cells from three independent experiments. **f** Cell cycle analysis of RPE1 cells assessed by flow cytometry at the indicated concentrations of ascorbic acid. **g** Quantification of the percentage of cells in G1, S and G2/M phases. $n = 3$ biological replicates for each concentration. **h** Time-laps imaging with bright-field microscopy for analysis of the total cell-cycle duration. Representative still images are shown. T = 0:00 is designated as mitotic cell rounding (time shown in h : min). Scale bar, 100 μm. **i**, Quantification of the total duration of cell cycle from **h**. The time from mitotic cell rounding to the next rounding was measured. $n > 30$ cells from independent three experiments. In **e**, **g**, and **i**, data are mean ± S.D., and P values were calculated by two-tailed unpaired Student's t-test. n.s.: not significant.

duration of mitosis compared to the condition with 3-min imaging intervals (Fig. 5b, c and Supplementary Movie 4). In addition, the cells that entered mitosis at later time points during the imaging showed longer mitotic durations in the short-interval condition, as seen in the high-light illumination condition with 3-min intervals (Figs. 2b, 5d). Remarkably, the addition of ascorbic acid to the imaging media restored the precise duration of mitosis (from 28.0 min to 21.2 min on average) even in the short-interval condition (Fig. 5b, c and Supplementary Movie 4). The delay of chromosome alignment was also alleviated in the presence of 500 μM ascorbic acid (Fig. 1e and Supplementary Fig. 5a, b) This outstanding photoprotective property of ascorbic acid was observed throughout the entire time-lapse acquisition (~12 h), abolishing the effect of increasing mitotic duration with accumulating amount of blue-light exposure (Fig. 5d and Supplementary Fig. 5c). In addition, the efficacy of ascorbic acid in preventing the prolongation of mitosis during the short-interval imaging was further validated in the cells expressing TUBB5-mNG and H2B-mScarlet (Supplementary Fig. 5d-f). Thus, these data indicate that the application of ascorbic acid enables to capture the accurate mitotic processes at very high temporal resolution in fluorescence live imaging.

## Discussion
In this study, we demonstrate that a submillimolar concentration of ascorbic acid can be safely applied to reduce phototoxicity-induced mitotic defects during fluorescence live cell imaging. We first analyzed the phototoxicity of excitation light on mitotic dynamics and observed that, consistent with a previous report, the duration of mitosis was significantly prolonged by high dose illumination with excitation light of 488 nm. In particular, the process of chromosome alignment at the metaphase plate was delayed. Blue light illumination has been shown to induce ROS-mediated DNA damage, such as oxidation of DNA bases and DNA double-strand breaks (DSBs)[26,27]. Moreover, DSBs generated during mitosis are reported to cause the delay of chromosome alignment and the activation of spindle assembly checkpoint, resulting in prolongation of the total mitotic duration[28–30]. Similarly, the light-induced prolonged mitosis observed under our experimental conditions may be attributed to DSBs induced by high-light illumination. Besides mitotic duration and chromosome alignment, our analysis additionally revealed that the timing of centrosome separation is another photosensitive process in mitosis. Centrosome separation is regulated by a mitotic kinase cascade consisting of PLK1, NEK2, and others[31]. Previous reports have shown that DNA damage inactivates these mitotic kinases leading to the inhibition of centrosome separation[32,33]. Thus, the light-induced DNA damage is likely to be a common cause of the here observed mitotic defects in fluorescence live cell imaging.

To alleviate the phototoxicity to mitosis in fluorescence live cell imaging, we examined the effect of reduced light exposure prior to entering mitosis by introducing a cell-cycle synchronization step before the start of the acquisition experiment. Although two different synchronization methods allowed shortening of the total light exposure time before mitosis, these approaches did not suppress the light-induced mitotic prolongation, likely due to cytotoxic side-effects. Aphidicolin, used in the two-step synchronization method, is known to cause replication stress even at low concentrations[34,35]. Cells with mild replication stress can escape from the S phase checkpoint and progress into mitosis, although they tend to exhibit prolonged mitosis[36]. Therefore, the cytotoxic effect of aphidicolin may amplify the phototoxic effect instead of relieving it. Even in the case of synchronization with palbociclib treatment, replication stress is a non-negligible factor. Long-term treatment of palbociclib has been shown to induce replication stress by downregulating the components of the replisome[37]. While 24 h of palbociclib treatment is not likely sufficient to induce obvious replication stress, it is possible that mild replication stress by palbociclib treatment, together with high-light illumination, synergistically induces mitotic prolongation. Thus, our work suggests that the approaches of cell-cycle synchronization are not suitable for alleviating the light-induced mitotic prolongation in fluorescence live cell imaging due to their known activation of replication stress.

Our antioxidant screen identified ascorbic acid as a potent supplement that can be added to the imaging media to suppress light-induced mitotic prolongation. In contrast, other antioxidants known to prevent ROS activity and cellular phototoxicity did not alleviate the mitotic phototoxicity in this study. One possible explanation for the notable impact of ascorbic acid could be its strong scavenging activity against a specific type of ROS. Several studies have shown that ascorbic acid has a markedly superior ability to neutralize superoxide radicals, a major component of ROS, in comparison to other antioxidants that were tested in our screen[38–40]. In addition, ascorbic acid has been observed to possess an additional property of compacting the higher-order structure of DNA. This phenomenon arises from its direct interaction with DNA, leading to the suppression of photo-induced double-strand breaks (DSBs) formation[41,42]. Therefore, these unique properties of ascorbic acid might be responsible for its outstanding effect in alleviating the light-induced mitotic prolongation. Taken together, this study revealed ascorbic acid as the optimal antioxidant to reduce the mitotic phototoxicity occurring during prolonged light exposure, and thus offers a practical solution for low-phototoxicity fluorescence live cell imaging with very high temporal resolution.

## Methods
**Cell culture and cell-cycle synchronization.** RPE1 cells were grown in Dulbecco's Modified Eagle's Medium F-12 (DMEM/F-12) (Nacalai Tesque, 11581-15) with 10% FBS (NICHIREI, 175012) and 1% penicillin/streptomycin (Nacalai Tesque, 09367-34). BJ-5ta cells were grown in a 4:1 mixture of DMEM (Nacalai Tesque,

**a**

Short-interval condition

| Light condition | Excitation | Laser power | Exposure time | Z-stack | Intervals |
|---|---|---|---|---|---|
| low | 488 nm | 15% (0.70 W/cm²) | 50 msec | 1 μm step × 21 slices | 30 sec |
| | 561 nm | 25% (1.14 W/cm²) | 75 msec | | |

**b**

Control condition (3-min intervals)

Ascorbic acid 0 μM

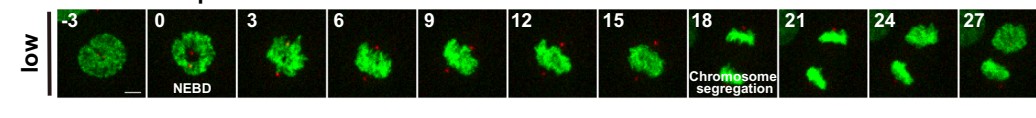

Short-interval condition (30-sec intervals)

Ascorbic acid 0 μM

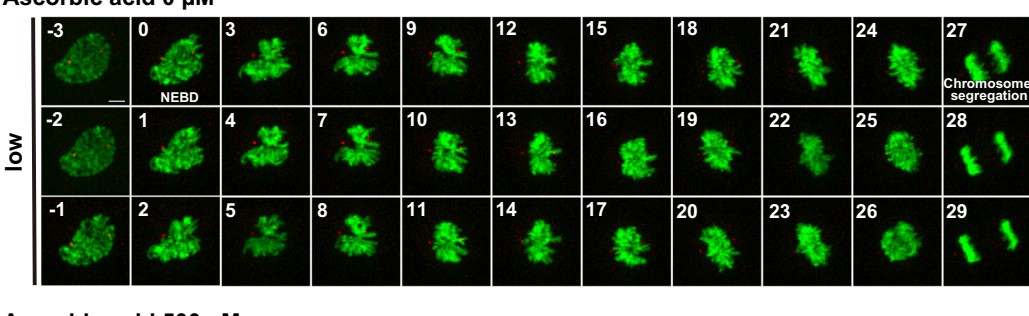

Ascorbic acid 500 μM

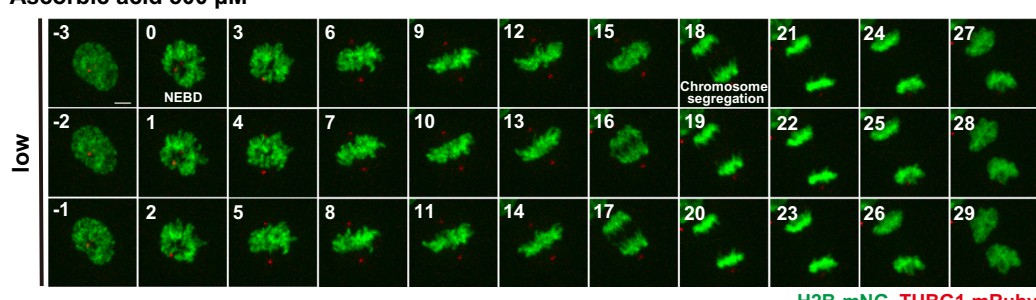

H2B-mNG, TUBG1-mRuby2

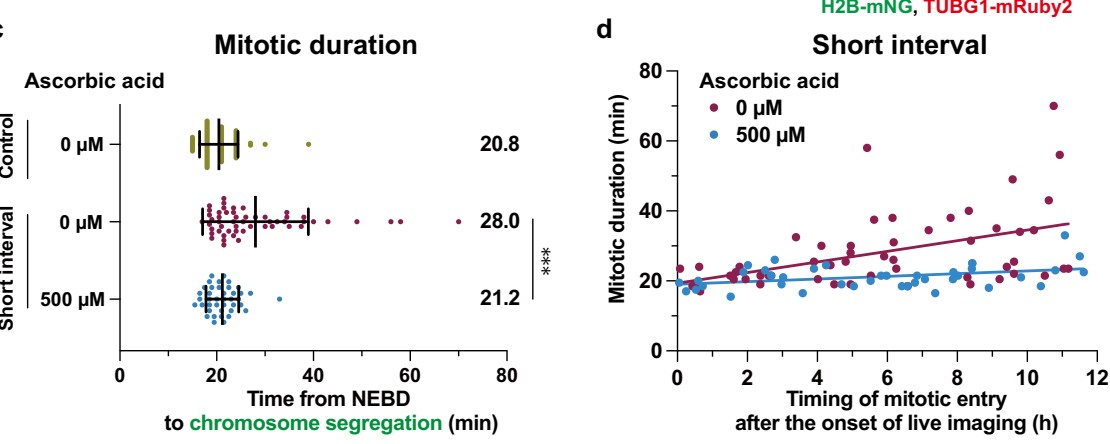

**Fig. 5 The addition of ascorbic acid to imaging buffer enables very high temporal resolution imaging of mitosis without obvious photodamage. a** The imaging condition used for live cell imaging with 30-sec intervals. **b** Time-lapse imaging of mitotic cells in the indicated conditions. Representative still images with different settings of brightness and contrast are shown. T = 0 is designated as NEBD (time shown in min). Scale bar, 5 μm. **c** Quantification of mitotic duration from **b**. The time from NEBD to chromosome segregation was measured. $n \geq 40$ cells from four independent experiments. Data are mean ± S.D., and $P$ values were calculated by Mann-Whitney $U$-test. ***$P < 0.001$. **d** The correlation between the timing of mitotic entry (NEBD) after the start of live imaging (x-axis) and mitotic duration (y-axis) in the short-interval condition from **b**. The regression lines for the indicated conditions are shown.

08459-64) and Medium 199 (ThermoFisher, 11150059) with 10% FBS and 1% penicillin/streptomycin. All cell lines were cultured at 37 °C in a humidified 5% $CO_2$ incubator. For cell-cycle synchronization with palbociclib, RPE1 cells were treated with 300 nM palbociclib (Selleck, S1116) for 24 h and released by replacing with fresh media. Palbociclib stock solution (100 μM) was prepared in DMSO and added 3 μL to 1 mL of fresh media. For the starvation and aphidicolin synchronization, cells were first serum starved with medium containing 0.1% FBS for 24 h and subsequently released by replacing with medium containing 20% FBS. After 4 h of the release, cells were treated with 1 μM aphidicolin (Sigma-Aldrich, A0781) for 18 h and released by replacing with fresh media. Aphidicolin stock solution (1 mg/mL) was prepared in DMSO and added 0.34 μL to 1 mL of fresh media.

**Generation of stable cell lines**. To generate stable cell lines that constitutively express fluorescent-protein fusions of organelle markers, a retroviral-based gene transfer system was utilized with the following constructs in accordance with the manufacturer's protocol (retroviral gene transfer and expression user manual, Clontech): pQCXIZ-H2B-mNeonGreen (chromosome), pQCXIZ-H2B-mScarlet-i (chromosome), pQCXIZ-γ-tubulin-mRuby2 (centrosome), and pQCXIZ-TUBB5-mNeonGreen (microtubule). Stable cells with low expression of the constructs were sorted using a BD FACS Aria III (Becton Dickinson).

**Antioxidants**. The following antioxidants were used in this study: Trolox (Vector Laboratories, CB-1000-2), Zeaxanthin (Santa Cruz, sc-205544), L-Sodium Pyruvate (Nacalai Tesque, 06977-34), α-tocopherol (Nacalai Tesque, 34114-54), Rutin (Nacalai Tesque, 30319-04), N-Acetyl-L-cysteine (Nacalai Tesque, 00512-84) and Ascorbic acid (Nacalai Tesque, 03420-52). Ascorbic acid (100 mM) was prepared in water and stored at −20 °C. Zeaxanthin (2 mM), α-tocopherol (10 mM), Rutin (10 mM) and NAC (100 mM) were prepared in DMSO and stored at −20 °C. Trolox (100 mM) and Sodium Pyruvate (100 mM) were stored at 4 °C. Each antioxidant was diluted in 500 μL of the culture medium at a final concentration before live cell imaging.

**Live cell imaging**. For fluorescence live cell imaging, cells stably expressing fluorescent-protein fusions of organelle markers were cultured in 35-mm glass-bottom dishes (Greiner-bio-one, #627870) at 37 °C in a 5% $CO_2$ atmosphere. The growth medium was replaced with a modified medium containing each antioxidant 3 hr before live cell imaging. A spinning disk confocal scanner box, the CellVoyager CV1000 (Yokogawa Electric Corp) equipped with a 40× oil-immersion objective and a back-illuminated EMCCD camera was used for live cell imaging. The excitation lasers at 488 nm (20 mW, diode laser) and 561 nm (20 mW, solid-state laser) were used. Images were acquired as z-stacks with a 40 × 1.30 NA oil-immersion lens every 3 min or 30 sec for 12 h. Maximum intensity projections of representative images were created using Fiji. The power of excitation light on a 40× oil-immersion objective lens was measured manually by using laser power meter 3664 (HIOKI, 3664), a handheld optical power meter. For analysis of mitotic events, centrosome separation was defined by the occurrence of an inter-centrosomal distance greater than 4 μm, NEBD was determined by the changes in the morphology of the nucleus, and chromosome segregation was judged by the timing at which the chromosomes were completely separated.

For analysis of the total duration of cell cycle, RPE1 cells were cultured in a 24-well SensoPlate (Greiner-bio-one, #662892) at 37 °C in a 5% $CO_2$ atmosphere. The growth medium was replaced

with the one containing 500 μM of ascorbic acid 3 h before live cell imaging. A confocal scanner box, CellVoyager CQ1 (Yokogawa Electric Corp) equipped with an sCMOS camera was used for live cell imaging. Images were acquired with a 10× objective lens every 10 min for 48 h.

**MTT assay**. RPE1 cells were seeded at a density of $1 \times 10^4$ cells/mL in a 96-well plate (WATSON, 197-96CIPS). After 24 h, the medium was replaced with fresh one containing each concentration of ascorbic acid. The plate was then incubated for 72 h, and the medium was replaced with the one containing MTT (Thermo-Fisher, 0.5 mg/mL final concentration). After culturing for 3 h, the medium was discarded and MTT formazan crystals were dissolved by adding isopropanol. The optical density was read at 600 nm using FLUOstar OPTIMA (BMG LABTECH). Cell viability was calculated by the normalization of optical density to the control treatment. Images were acquired with bright-field (50-msec exposure) using CellVoyager CQ1 (Yokogawa Electric Corp) equipped with a 10× objective lens and an sCMOS camera.

**Measurement of medium pH**. For measurement of medium pH, RPE1 cells were cultured in a 24-well plate (WATSON, 197-24CS). Medium was replaced with fresh one containing 500 μM of ascorbic acid 3 h before measurement. Medium pH was measured using Twin pH Meter II LQUAtwin (AS ONE, As-pH-22) at 37 °C in a 5% $CO_2$ atmosphere. Data were acquired from 3 biological replicates.

**Flow cytometry**. For cell-cycle analysis, RPE1 cells were seeded at a density of $1 \times 10^4$ cells/mL in a 6-well plate (WATSON, 197-06CPS). One day later, the medium was replaced with the one containing 500 μM of ascorbic acid. After 24 h incubation, the cells were trypsinized with 0.5 g/L-Trypsin / 0.53mmol/L-EDTA Solution (Nacali Tesque, 32778-05) for 3 min and transferred to a 15 mL tube, fixed in ice-cold 70% Ethanol, and frozen at −20 °C overnight. Fixed cells were pelleted and washed three times in PBS. 200 μL of Muse Cell Cycle Reagent (Luminex, MCH100106) was added to the cell pellet and incubated for 30 min at room temperature in the dark. Data were acquired by a compact flow cytometry, Muse cell analyzer (Luminex). Gates and regions were placed around populations of cells with appropriate size. The DNA content was analyzed using FlowJo software (BD Biosciences).

**Statistics and Reproducibility**. Statistical comparison between the data from different groups was performed in PRISM v.9 software (GraphPad) using Mann-Whitney *U*-test or two-tailed unpaired Student's t-test. P values < 0.05 were considered statistically significant. All data shown are mean ± S.D. The sample size is indicated in the figure legends.

**Reporting summary**. Further information on research design is available in the Nature Portfolio Reporting Summary linked to this article.

## Data availability
All data supporting the findings of this study are available from the corresponding authors on reasonable request. The source data behind the graphs in the paper can be found in Supplementary Data 1.

## Code availability
The FIJI or ImageJ macros used in this study are available from the corresponding authors on request.

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

## Acknowledgements

We thank Mariya Genova for the proofreading of the manuscript, Dr. Yuji Ikegaya at Graduate School of Pharmaceutical Sciences at the University of Tokyo and Dr. Hiroshi Nishina at Medical Research Institute at Tokyo Medical and Dental University for sharing equipment, and the Kitagawa lab members for technical supports and helpful discussions. This work was supported by JSPS KAKENHI grants (Grant numbers: 18K06246, 19H05651, 20K15987, 20K22701, 21H02623, 22H02629) from the Ministry of Education, Science, Sports and Culture of Japan, the PRESTO program (JPMJPR21EC) of the Japan Science and Technology Agency, Takeda Science Foundation, The Uehara Memorial Foundation, The Research Foundation for Pharmaceutical Sciences, Koyanagi Zaidan, The Kanae Foundation for the Promotion of Medical Science, Kato Memorial Bioscience Foundation, and Tokyo Foundation for Pharmaceutical Sciences.

## Author contributions

S.H. conceived and designed the study. T.H. performed most of the experiments. R.T. performed cell-cycle analysis. T.K. established a cell line for fluorescence imaging. M.F. and T.C. provided suggestions. T.H., S.H., and D.K. analyzed the data. T.H., S.H., and D.K. wrote the manuscript. All authors contributed to discussions and manuscript preparation.

## Competing interests

The authors declare no competing interests.
