## [Peer Review File · Communications Biology]

Reviewers' comments:

Reviewer #1 (Remarks to the Author):

The issue of phototoxicity in the fluorescence imaging of live cells and the use of antioxidants to alleviate this problem are enduring but not new issues. That said, time lapse fluorescence imaging, particularly with 488 nm excitation, is an important discipline in basic cell biological research. Any studies on phototoxic reactions to high intensity light that document sub-lethal cellular dysfunction and present realistic means to alleviate this problem represent important contributions to the field. This report by Harada et al. successfully provides both. Thus, I recommend that, pending the authors' addressing any concerns of reviewers, this paper be put on track for acceptance.

Overall, the manuscript is well written and the observations well documented. I find no major structural defects. Below I outline a few issues to be addressed.

1. Page 6, lines 22-25 in reference to data shown in figure 2: The authors compare mitosis duration for cells that come into mitosis within 6 hours of the onset of imaging as compared to those that enter mitosis 6-12 hours after the onset of imaging. This is readily understandable. Less clear and initially confusing is the statement that mitosis is prolonged for "even those in the total imaging time (0-12 hr)" implying that this is a separate category. This category appears to be the 0-6 hr and 6-12 hr data taken together. If this is the case, the authors would do well to say so.

2. The authors do not mention what effect, if any, their concentrations of ascorbic acid have on the pH of the culture medium. One would expect that the HEPES in the medium should buffer adequately but an empirical test is easy and would address a loose end for some readers.

3. The authors appropriately test whether the concentration of ascorbic acid used has any generally harmful effects on the cells (Figure 4). To this end they primarily use the MTT assay which is a measure of metabolic activity and flow cytometry analysis of the proportion of cells in various cell cycle stages. The results of these assays are appropriate to be included in the study and well support the case that ascorbic acid does not have adverse effects at moderate concentrations. However, in the context of a study on the timing of mitotic events, these assays are indirect. I think that the authors could readily test and perhaps bolster their case by characterizing the timing of mitotic events in the presence and absence of moderate concentrations of ascorbic acid without fluorescence excitation. This can be done by time lapse filming of cells with phase contrast microscopy in green light. This has the added benefit that the authors could characterize the effects of ascorbic acid, if any, on the overall cell cycle duration – say NEB to NEB. Regardless of outcome, this would be a useful contribution. For full cell cycle analysis, I suggest that the authors use a ~10X objective to provide a large field of view that will accommodate the motility of RPE1 cells.

Reviewer #2 (Remarks to the Author):

The fluorescence microscopy-based approaches are crucial for myriad aspects of cell biology. Changes that occur by excessive illumination can lead to artifacts and misbehavior of the cellular structures and cellular processes. In this manuscript, the authors have demonstrated that a submillimolar concentration of ascorbic acid can help to mitigate the impact of phototoxicity-induced damages during mitotic progression, thus suggesting that the use of ascorbic acid could be helpful for live-imaging analysis.

This manuscript by Harada et al. is interesting as the authors have shown that ascorbic acid can dampen phototoxicity's impact by a 488 nm laser during imaging. The major weakness is the work does not explain why ascorbic acid, but not other compounds rescues the phototoxicity-based mitotic delay. Further, the authors have used only one single protein tagged with mNG to study the impact of phototoxicity-mediated mitotic delay, which does not readily explain if phototoxicity induced by other fluorophores (e.g., mCherry) is also dampened using ascorbic acid (see my major points).

Major points:

1. As mentioned above, it remained unclear how ascorbic acid reduces the impact of phototoxicity. Why did other tested compounds fail to minimize the effect of phototoxicity by 488 nm laser?
2. Do the authors know if mCherry-H2B expressing cells image with a time-frame of 30s also significantly delays mitotic progression? And, if yes, does the addition of ascorbic acid rescue these delays?
3. It appears to me that the cells treated with ascorbic acid elongate mitotic spindles more than untreated cells (Figure 5). Can the authors quantify this phenotype and comment on it?
4. It needs to be clarified if the centrosome separation defect seen upon high laser exposure is mitigated by ascorbic acid treatment.
5. For a relatively stable protein H2B, the laser power (488nm) used by the authors under high and low conditions is pretty extreme (100% and 15%). Also, It would be nice to reproduce these phenotypes using a protein that localizes to other structures (kinetochore/centrosomes) during mitotic progression.

Reviewer #3 (Remarks to the Author):

This is a very well written and easy to read manuscript and it was a pleasure to review. The finding that ascorbic acid is a powerful antioxidant that minimizes phototoxic effects that impact mitotic processes will be valuable for the community.

I have several comments on metadata reporting that needs to be improved so this work could be repeated and some comments that should be discussed in the paper as well as suggestions for future work.

1) I am inclined to think that because the ascorbic acid localizes to the DNA in the cells it is in close proximity to the H2B-mNG and ready to neutralize any reactive oxygen species. Perhaps that is why it is uniquely capable of reducing phototoxicity in these experiments. It would add to the paper if they authors discussed this kind of localization of the ROS in the cell in the discussion section of the paper. This could also explain why Trolox didn't have the expected impact on phototoxicity.

I think this paper is comprehensive and tells a complete story with all the necessary controls but it would be interesting to see if ascorbic acid has the same unique impact when compared to the other antioxidants if the mNG is fused to a protein that has a different localization such as a cytosolic protein.

2) Again, I don't think this is required for publication of this article but it would be good to show a similar reduction in phototoxicity with ascorbic acid and correct mitotic processes with different cell lines.

3) Page 6, line 4, the statement about timing and low and high light conditions was confusing since you are talking about the time before NEBD. It would be helpful to put the negative sign before the times to make it clearer. At first, I thought there was a mistake and a reversal of the time data for the two conditions.

4) The cell culture and cell-cycle synchronization section requires more details. Add company and

catalog numbers for all reagents. Add the solvents for Palbociclib and aphidicolin solutions and details about stock solution concentrations and how much was added to what volume of media so these experiments can be easily reproduced.

5) Antioxidants section: add catalog numbers for all reagents, add details of the concentration of the stock solutions and the volume of stock solution added to what volume of media.

6) Fluorescence live cell imaging: provide company and model of EMCCD, add exposure time details for both images, add the catalogue number for the power meter, add details of the power sensor that was used – was it a slide format or a sensor somehow mounted on the microscope stage, describe how the laser power was measured – was it with immersion media and a slide-based sensor that can be used with immersion media or was it done on a lower magnification air objective lens? Provide information about the laser – make, model, power, dichroic and emission filter specifications, environmental control system.

7) MTT Assay: Provide company and catalog number for 96-well plate, provide company and product information for HCA system, details of 10x lens, type of sCMOS camera and exposure time for images.

8) Flow cytometry: add company and catalog number for 6-well plate, detail trypsin methods – company and catalog number, concentration and time incubated with trypsin, how long were the cells in ice-cold ethanol for? What type of container were the cells fixed in? how were cells washed? Add catalog number for muse cell cycle reagent, what is it dissolved in? Provide details on the cell analyzer and a workflow for cell analysis with FlowJo.

9) It is difficult to properly annotate and share image data but this reviewer strongly encourages the authors to make their code available on GitHub.

10) Was an imaging facility and/or their expertise used for any part of these experiments? If so they should be acknowledged.

11) For intensity projections for the High illumination conditions were the images saturated or is that just how they are displayed in the figures?

12) The figure legends are very comprehensive but more details on how the time of NEBD, centrosome separation and chromosome segregation were identified is needed in the methods section. Even if this was done by visual inspection what characteristics within the images were used to identify these stages?

13) For the Nature Portfolio report summary – it is indicated that antibodies were used in the study but then later it states no antibodies were used. It is recommended to authenticate cell lines. How often were cells tested for mycoplasma?

14) The statistical methods used and quantitative data presentations in the manuscript are appropriate.

Reviewer #1 (Remarks to the Author):

The issue of phototoxicity in the fluorescence imaging of live cells and the use of antioxidants to alleviate this problem are enduring but not new issues. That said, time lapse fluorescence imaging, particularly with 488 nm excitation, is an important discipline in basic cell biological research. Any studies on phototoxic reactions to high intensity light that document sub-lethal cellular dysfunction and present realistic means to alleviate this problem represent important contributions to the field. This report by Harada et al. successfully provides both. Thus, I recommend that, pending the authors' addressing any concerns of reviewers, this paper be put on track for acceptance.

Overall, the manuscript is well written and the observations well documented. I find no major structural defects. Below I outline a few issues to be addressed.

We are deeply grateful to the reviewer for taking the time to read our manuscript and providing constructive comments. The following are point-by-point responses to the questions.

1. Page 6, lines 22-25 in reference to data shown in figure 2: The authors compare mitosis duration for cells that come into mitosis within 6 hours of the onset of imaging as compared to those that enter mitosis 6-12 hours after the onset of imaging. This is readily understandable. Less clear and initially confusing is the statement that mitosis is prolonged for “even those in the total imaging time (0-12 hr)” implying that this is a separate category. This category appears to be the 0-6 hr and 6-12 hr data taken together. If this is the case, the authors would do well to say so.

We thank the reviewer for this comment. As pointed out, “the total imaging time” refers to the combined data taken during the 0-6 hr and 6-12 hr periods, and thus we have modified the manuscript (page 6, line 25 in the revised manuscript) and the figure to make it understandable.

2. The authors do not mention what effect, if any, their concentrations of ascorbic acid have on the pH of the culture medium. One would expect that the HEPES in the medium should buffer adequately but an empirical test is easy and would address a loose end for some readers.

We thank the reviewer for this suggestion. To test the ascorbic acid's impact on pH, we have measured the pH of the media containing ascorbic acid inside the 5% CO₂ incubator. We confirmed the pH values were nearly identical in the presence or absence of 500 μ M ascorbic acid (pH 7.97 \pm 0.01 and pH 7.94 \pm 0.02, respectively). This data indicates that the influence of ascorbic acid on pH at a moderate concentration can be sufficiently buffered under the cell culture condition. We have added this information in the revised manuscript (page 9, lines 21-24 in the revised manuscript).

3. The authors appropriately test whether the concentration of ascorbic acid used has any generally harmful effects on the cells (Figure 4). To this end they primarily use the MTT assay which is a measure of metabolic activity and flow cytometry analysis of the proportion of cells in various cell cycle stages. The results of these assays are appropriate to be included in the study and well support the case that ascorbic acid does not have adverse effects at moderate concentrations. However, in the context of a study on the timing of mitotic events, these assays are indirect. I think that the authors could readily test and perhaps bolster their case by characterizing the timing of mitotic events in the presence and absence of moderate concentrations of ascorbic acid without fluorescence excitation. This can be done by time lapse filming of cells with phase contrast microscopy in green light. This has the added benefit that the authors could characterize the effects of ascorbic acid, if any, on the overall cell cycle duration – say NEB to NEB. Regardless of outcome, this would be a useful contribution. For full cell cycle analysis, I suggest that the authors use a \sim 10X objective to provide a large field of view that will accommodate the motility of RPE1 cells.

We thank the reviewer for this important suggestion. As requested by the reviewer, we have analyzed the total duration of cell cycle (from mitosis to mitosis) in both the presence and absence of ascorbic acid. Live imaging with bright-field microscopy revealed that there was no significant difference in the total cell cycle duration between conditions with and without 500 μ M ascorbic acid (17.5 hr and 17.9 hr on average, respectively, new data in Fig. 4h, i). This result suggests that, when considered along with the data on the proportion of cell-cycle phases (Fig. 4f, g), the optimal concentration of ascorbic acid does not affect the timing of mitotic events. We have added the new data and modified the result section accordingly (page 9, lines 30-31 in the revised manuscript).

Reviewer #2 (Remarks to the Author):

The fluorescence microscopy-based approaches are crucial for myriad aspects of cell biology.

Changes that occur by excessive illumination can lead to artifacts and misbehavior of the cellular structures and cellular processes. In this manuscript, the authors have demonstrated that a submillimolar concentration of ascorbic acid can help to mitigate the impact of phototoxicity-induced damages during mitotic progression, thus suggesting that the use of ascorbic acid could be helpful for live-imaging analysis.

This manuscript by Harada et al. is interesting as the authors have shown that ascorbic acid can dampen phototoxicity's impact by a 488 nm laser during imaging. The major weakness is the work does not explain why ascorbic acid, but not other compounds rescues the phototoxicity-based mitotic delay. Further, the authors have used only one single protein tagged with mNG to study the impact of phototoxicity-mediated mitotic delay, which does not readily explain if phototoxicity induced by other fluorophores (e.g., mCherry) is also dampened using ascorbic acid (see my major points).

We appreciate the reviewer for taking the time to critically read our manuscript for providing insightful feedback. We agree that the main points the reviewer has mentioned above are critical for our manuscript. The following are point-by-point responses to the questions.

Major points:

1. As mentioned above, it remained unclear how ascorbic acid reduces the impact of phototoxicity. Why did other tested compounds fail to minimize the effect of phototoxicity by 488 nm laser?

We thank the reviewer for this important comment. Given that reactive oxygen species (ROS) generation is the primary cause of phototoxicity, one possible answer is that the neutralization of ROS by ascorbic acid could be much greater than other compounds tested in our screen. To test this hypothesis, we tried to measure ROS levels and compare the ROS scavenging ability among these antioxidants in fluorescent live imaging by using fluorogenic probes that can detect ROS molecules. Although we tried several different probes (CellROX Green, Invitrogen; CellROX Orange, Invitrogen; ROS-ID Total ROS/Superoxide detection kit, Enzo Life Sciences), it was hard to evaluate ROS levels quantitatively because these probes exhibited photobleaching under the condition of high-light illumination. Therefore, instead of experimental approaches, we have discussed a specific ROS scavenging activity of ascorbic acid in the revised manuscript. In previous research, ascorbic acid has been shown to possess a superior ability to neutralize superoxide radicals, a major component of ROS, compared to other compounds tested in our screen (Mazor et al., 2006). The extent to which superoxide radicals cause mitotic phototoxicity in fluorescence live imaging is unclear, but this specific ability of ascorbic acid might result in reducing photodamage. Additionally, as written in the original manuscript, ascorbic acid has been found to have the property of preventing DNA double-strand breaks (Yoshikawa et al., 2006). These unique

properties of ascorbic acid might be responsible for the remarkable ability to reduce phototoxicity. This point has been added to the discussion section of the revised manuscript (page 13, lines 12-25).

2. Do the authors know if mCherry-H2B expressing cells image with a time-frame of 30s also significantly delays mitotic progression? And, if yes, does the addition of ascorbic acid rescue these delays?

We thank the reviewer for this comment. Live imaging with short intervals (30-sec intervals) leads to a high incidence of light exposure, and we have concerned about photobleaching of mCherry in the short-interval condition. Thus, we generated RPE1 cells expressing H2B-mScarlet, which is known as a more photostable protein than mCherry, along with TUBB5-mNG. In the short-interval condition (30-sec interval), the duration of mitosis of the cells expressing H2B-mScarlet was prolonged compared to the control condition (3-min interval). And this mitotic prolongation was significantly alleviated by the addition of ascorbic acid (new data in Fig. S4e, f, S3a, b). We have added the new data and modified the result section accordingly (page 10, lines 28-30 in the revised manuscript).

3. It appears to me that the cells treated with ascorbic acid elongate mitotic spindles more than untreated cells (Figure 5). Can the authors quantify this phenotype and comment on it?

We thank the reviewer for pointing this out. In response to the request, we quantified the spindle length of the metaphase cells and confirmed that there was no significant difference in the presence or absence of ascorbic acid ($10.7 \pm 1.01 \mu\text{m}$ or $10.8 \pm 1.00 \mu\text{m}$, respectively, in supplementary figure for reviewer). The spindle length often looks different in MIP (Maximum intensity projection) images due to the spatial orientation of the spindle. We think that this factor could affect the apparent shape of the spindle and it would not be the phenotype caused by ascorbic acid.

a, Metaphase of RPE1 cells expressing H2B-mNG and TUBG1-mRuby2 for indicated conditions in the low-light illumination. Scale bar; 10 μm . **b**, Quantification of spindle length from **a**. $n > 30$ cells from three independent experiments. In **b**, data are mean \pm S.D., and P values were calculated by Mann-Whitney U-test. n.s.: not significant.

4. It needs to be clarified if the centrosome separation defect seen upon high laser exposure is mitigated by ascorbic acid treatment.

We show the data that the light-induced delay of centrosome separation was suppressed by ascorbic acid in the original and the revised manuscripts (Fig. 3a, j).

5. For a relatively stable protein H2B, the laser power (488nm) used by the authors under high and low conditions is pretty extreme (100% and 15%). Also, It would be nice to reproduce these phenotypes using a protein that localizes to other structures (kinetochore/centrosomes) during mitotic progression.

We thank the reviewer's suggestion. To address the reviewer's request, we generated RPE1 cells that stably express mNG-fused TUBB5, localizing to microtubules, and H2B-mScarlet. Similar to the case of H2B-mNG, the treatment of 500 μ M ascorbic acid almost completely mitigated the prolongation of mitosis in RPE1 cells expressing TUBB5-mNG caused by high irradiation of blue light (new data in Fig. S3a, b). This data suggests that the photoprotective effect of ascorbic acid does not depend on the localization of fluorescent proteins. We have added the new data and modified the result section accordingly (page 8, line 23 - page 9, line 2 in the revised manuscript).

Reviewer #3 (Remarks to the Author):

This is a very well written and easy to read manuscript and it was a pleasure to review.

The finding that ascorbic acid is a powerful antioxidant that minimizes phototoxic effects that impact mitotic processes will be valuable for the community.

I have several comments on metadata reporting that needs to be improved so this work could be repeated and some comments that should be discussed in the paper as well as suggestions for future work.

We are sincerely grateful to the reviewer for taking the time to read our manuscript and providing helpful feedback. The following are point-by-point responses to the questions.

1) I am inclined to think that because the ascorbic acid localizes to the DNA in the cells it is in close proximity to the H2B-mNG and ready to neutralize any reactive oxygen species. Perhaps that is why it is uniquely capable of reducing phototoxicity in these experiments. It would add to the paper if they authors discussed this kind of localization of the ROS in the cell in the discussion section of the paper. This could also explain why Trolox didn't have the expected impact on phototoxicity.

I think this paper is comprehensive and tells a complete story with all the necessary controls but it would be interesting to see if ascorbic acid has the same unique impact when compared to the other antioxidants if the mNG is fused to a protein that has a different localization such as a cytosolic protein.

We thank the reviewer for the insightful comment. Indeed, it is known that fluorescent proteins generate ROS and damage cellular components in their vicinity (Takemoto et al., 2013). To address the reviewer's inquiry regarding whether ascorbic acid exhibits a photoprotective effect when the mNG is fused to a cytosolic protein, we generated RPE1 cells that stably express mNG-fused TUBB5, localizing to microtubules, along with mScarlet-H2B. We found that ascorbic acid mitigated the prolongation of mitosis in RPE1 cells expressing TUBB5-mNG caused by high amounts of blue light (new data in Fig. S3a, b). This suggests that the significant impact of ascorbic acid in preventing mitotic phototoxicity is not influenced by the localization of fluorescent proteins. We have added the new data and modified the result section accordingly (page 8, line 23 - page 9, line 2 in the revised manuscript).

2) Again, I don't think this is required for publication of this article but it would be good to show a similar reduction in phototoxicity with ascorbic acid and correct mitotic processes with different cell lines.

We thank the reviewer for this comment. As requested by this reviewer, we verified the photoprotective effect of ascorbic acid in BJ-5ta which is a human diploid fibroblast cell line. As in the case of RPE1 cells, BJ-5ta cells also showed prolonged mitosis by the high illumination of blue light, and the treatment of ascorbic acid significantly alleviated the mitotic prolongation (new data in Fig. S3c, d, e). We have added the new data and modified the result section accordingly (page 9, lines 3-8 in the revised manuscript).

3) Page 6, line 4, the statement about timing and low and high light conditions was confusing since you are talking about the time before NEBD. It would be helpful to put the negative sign before the times to make it clearer. At first, I thought there was a mistake and a reversal of the time data for the two conditions.

We have added the negative sign before the time and modified the manuscript in the results section to avoid misunderstanding (page 6, line 4 in the revised manuscript).

4) The cell culture and cell-cycle synchronization section requires more details. Add company and catalog numbers for all reagents. Add the solvents for Palbociclib and aphidicolin solutions and details about stock solution concentrations and how much was added to what volume of media so

these experiments can be easily reproduced.

We have provided the details of the reagents in the Material and Method section (page 14 in the revised manuscript).

5) Antioxidants section: add catalog numbers for all reagents, add details of the concentration of the stock solutions and the volume of stock solution added to what volume of media.

We have added the details of antioxidants in the Material and Method section (pages 14-15 in the revised manuscript).

6) Fluorescence live cell imaging: provide company and model of EMCCD, add exposure time details for both images, add the catalogue number for the power meter, add details of the power sensor that was used – was it a slide format or a sensor somehow mounted on the microscope stage, describe how the laser power was measured – was it with immersion media and a slide-based sensor that can be used with immersion media or was it done on a lower magnification air objective lens? Provide information about the laser – make, model, power, dichroic and emission filter specifications, environmental control system.

We have provided additional information on EMCCD, power meter, and laser in the Material and Method section (page 15). The company and model of EMCCD and information of the laser were not available because these details were confidential in Yokogawa Electric Corp. Exposure time of each image is listed in Fig. 1a, 5a, S3c, S4d.

7) MTT Assay: Provide company and catalog number for 96-well plate, provide company and product information for HCA system, details of 10x lens, type of sCMOS camera and exposure time for images.

We have added the details about MTT assay in the Material and Method section (page 16). We have removed the mention of HCA system, as it was erroneously included and, it was not used. Information about the specific type of sCMOS camera was not obtained because of confidentiality reasons.

8) Flow cytometry: add company and catalog number for 6-well plate, detail trypsin methods – company and catalog number, concentration and time incubated with trypsin, how long were the cells in ice-cold ethanol for? What type of container were the cells fixed in? how were cells washed? Add catalog number for Muse cell cycle reagent, what is it dissolved in? Provide details on the cell analyzer and a workflow for cell analysis with FlowJo.

Following the suggestions from the reviewer, we have added the details of Flow cytometry in the Material and Method section (pages 16-17). The solvent of muse cell cycle reagent was not obtained because it is not open to the public.

9) It is difficult to properly annotate and share image data but this reviewer strongly encourages the authors to make their code available on GitHub.

The code used in this study is only the one to assemble MIP images into movies, which contains nothing special. We believe that providing it upon request would be beneficial.

10) Was an imaging facility and/or their expertise used for any part of these experiments? If so they should be acknowledged.

We performed all the experiments and analyses for ourselves.

11) For intensity projections for the High illumination conditions were the images saturated or is that just how they are displayed in the figures?

Images of the green channel under the high-illumination conditions are saturated, and representative images are shown with different settings of brightness and contrast to better visualize the red channel. We have added a note about it in the Figure legends.

12) The figure legends are very comprehensive but more details on how the time of NEBD, centrosome separation and chromosome segregation were identified is needed in the methods section. Even if this was done by visual inspection what characteristics within the images were used to identify these stages?

We thank the reviewer for this comment. For analysis of mitotic cells, NEBD was determined by the changes in the morphology of the nucleus, centrosome separation was defined by the inter-centrosomal distance ($>4 \mu\text{m}$), and chromosome segregation was judged by the timing when the chromosomes were completely separated. We have added these details in the Material and Methods section (page 15).

13) For the Nature Portfolio report summary – it is indicated that antibodies were used in the study but then later it states no antibodies were use. It is recommended to authenticate cell lines. How often were cells tested for mycoplasma?

We have corrected the statement about the use of antibodies and added information about cell lines in reporting summary.

14) The statistical methods used and quantitative data presentations in the manuscript are appropriate.

We thank the reviewer for careful and elaborate peer review.

REVIEWERS' COMMENTS:

Reviewer #1 (Remarks to the Author):

The authors have done a fine job in addressing my concerns in my original review. From my standpoint the manuscript is ready for acceptance for publication.

Reviewer #2 (Remarks to the Author):

The authors have fully addressed my concerns; thus, I feel the manuscript is ready for publication.

Reviewer #3 (Remarks to the Author):

The authors did an excellent job responding to the reviews. The additional details and additional experiments and discussion in the manuscript have really improved it. Well done.

Just one comment to add details about the sCMOS camera company and brand.